# OpenReview forum: "Circumventing Safety Alignment in Large Language Models via Embedding Space Toxicity Attenuation"
_ICLR.cc/2026/Conference — Submitted to ICLR 2026_

### Official Review · Reviewer_h6QN · 2025-10-24

**Soundness:** 3
**Presentation:** 3
**Contribution:** 3
**Rating:** 6
**Confidence:** 3

**Summary:**

This paper presents a method to bypass the safety guardrail of LLMs by manipulating the embedding space. The experiments on one harmful dataset show that the proposed method can achieve good attack performance on white-box models.

**Strengths:**

1. The proposed method is novel to my knowledge.

2. The attack performance is high.

3. The paper writing is good.

4. Many failure cases are provided.

**Weaknesses:**

1. If my understanding is correct, the proposed method is designed for white-box attack, which is a relatively easy task. Many simple but effective methods have been proposed for white-box attack, such as harmful finetuning (Fine-tuning aligned language models compromises safety, even when users do not intend to!). What is the advantage of the proposed method compared with these existing white-box attack methods?

2. Manipulating the embedding space for jailbreaking has been explored in existing studies, such as the SCAV method (Uncovering Safety Risks of Large Language Models through Concept Activation Vector). The difference lies in the specific embedding space to manipulate. In SCAV, they manipulate the embedding space in hidden Transformer layers while in this paper the authors select the token embedding space. Then a natural question is, what is the advantage of manipulating the token embedding space?

3. Only one harmful dataset is used for evaluation, which is not sufficient.

4. It is not clear what is the quality of the harmful response generated by the proposed method, since the proposed method leads to obvious quality decline in normal tasks.

5. More defense methods should be used in experiments to show that the proposed method can bypass existing defense methods.

6. More attack methods for white-box scenarios should be used for comparison. In this paper, most of the methods for comparison are for black-box setting, which is not fair.

**Questions:**

see my comments above.

---

> ### Author Response · Authors · 2025-11-29
>
> ***1. The advantage of ETTA compared with these existing white-box attack methods.***
>
> We agree that the advantage of ETTA can be more clearly clarified. Our experiments include representative white-box baselines such as Virus (fine-tuning method based on dual-goal optimization), Embedding Attack (GCG-style embedding poisoning), and COLD (white-box automated jailbreak generation with multi-dimension constraints). Compared with **Virus**, ETTA only learns a lightweight transformation from a small set which is far cheaper in time/memory rather than harmful fine-tuning. Virus is also weak on well-aligned models (e.g., 38.27% ASR on Llama-2 in our tests). Compared with Embedding Attack, the issue is not speed but stability. The gradient-optimized continuous prefix mainly forces a fixed affirmative prefix and often fails to steer later generations, so ASR is unstable and safety disclaimers reappear after the prefix, while ETTA induces a more consistent global jailbreak tendency. Compared with COLD, whose high ASR comes with high per-attack cost, ETTA achieves similar or better effectiveness without iterative gradient loops.
>
> ETTA’s main advantage is that it achieves strong jailbreaking with substantially lower overhead. Our core contribution is to demonstrate that modifying only the initial token embedding weights (before the first layer) suffices to bypass safety-alignment. We will tighten this positioning and make the efficiency–effectiveness comparison more explicit in the revision.
>
> ***2. The advantage of manipulating the token embedding space.***
>
> We agree that our work is conceptually related to prior embedding/representation manipulation approaches (including SCAV), while differing in where the intervention is applied.  The key advantage of manipulating the token embedding space is efficiency and cost. ETTA performs a **word-level pre-layer perturbation** on token embeddings, revealing a simpler, cost-effective control locus. Once the modified embeddings are constructed, the model runs normally without any layer-wise hooks or per-layer editing. In contrast, SCAV-style steering methods manipulating hidden-layer representations require **repeated interventions across layers**, which increases memory footprint and slows inference/training pipelines. To evaluate our work more comprehensively, we conducted a comparison with SCAV. Here is a brief assessment result (we use 100 test cases, on three 32-layer victim LLMs ):
>
> ASR:
> | Method | Llama-2-7B | Llama-3-8B | Mistral-7B |
> | ------ | ---------- | ---------- | ---------- |
> | ETTA   | 89%        | 85%        | 91%        |
> | SCAV   | 90%        | 88%        | 89%        |
>
> Timecost(s):
>
> | Method | Llama-2-7B | Llama-3-8B | Mistral-7B |
> | ------ | ---------- | ---------- | ---------- |
> | ETTA   | 30.9       | 24.1       | 31.3       |
> | SCAV   | 113.84     | 94.32      | 102.18     |
>
> Results indicate that ETTA achieves comparable attack effectiveness while requiring substantially less intervention overhead. We will include the full evaluation in the revision.
>
> ***3. Only one harmful dataset is used for evaluation, which is not sufficient.***
>
> We agree that relying on a single harmful dataset is relatively limiting, and have therefore added an additional benchmark. Specifically, we incorporate the *forbidden_question_set* from [1], which contains 390 harmful questions spanning 13 scenarios, and *JailbreakBench* from [2], which contains 100 harmful questions spanning 10 scenarios. Using the same experimental configuration as for AdvBench, we obtain the following ASR results:
>
> | Victim Model             | AdvBench | forbidden | JailbreakBench |
> | ------------------------ | -------- | --------- | -------------- |
> | Llama-2-7b-chat-hf       | 81.73%   | 68.72%    | 92.00%         |
> | Llama-3.2-3B-Instruct    | 84.81%   | 86.15%    | 95.00%         |
> | Mistral-7B-Instruct-v0.1 | 90.77%   | 77.44%    | 92.00%         |
>
> The performance patterns on *forbidden_question_set* and *JailbreakBench* are broadly consistent with those on AdvBench, which both supports the validity of AdvBench as an evaluation benchmark and indicates that our conclusions about ETTA’s relative effectiveness are robust across multiple harmful datasets.
>
> [1] "Do Anything Now": Characterizing and Evaluating In-The-Wild Jailbreak Prompts on Large Language Models, https://arxiv.org/abs/2308.03825
>
> [2] JailbreakBench: An Open Robustness Benchmark for Jailbreaking Large Language Models, https://arxiv.org/abs/2404.01318
>
> [3] A StrongREJECT for Empty Jailbreaks, https://arxiv.org/abs/2402.10260

---

> ### Author Response · Authors · 2025-11-29
>
> ***4. It is not clear what is the quality of the harmful response generated by the proposed method.***
>
> We thank the reviewer for raising this point. To better capture the semantic harmfulness of the generated outputs, we deliberately use a GPT4o-judge instead of HarmBench or LlamaGuard2 to judge ASR in our current evaluation.  As reported in [2], GPT4o achieves better *Agreement* and *False Positive Rate* than HarmBench and LlamaGuard2 according to expert annotations, which helps reduce false positives like superficial agreement or topic shifting.
>
> To further validate that our reported ASR indeed reflects genuinely harmful responses rather than artifacts of a specific judge, we conducted a preliminary experiment on Llama-2-7b-chat-hf using three independent evaluators: our GPT4o-judge, the JailbreakBench judge [2], and the StrongREJECT autograder [3]. The resulting ASRs are 81.73% (GPT4o-judge), 82.50% (JailbreakBench), and 76.35% (StrongREJECT), respectively. The close agreement between GPT4o-judge and JailbreakBench, and the still high ASR under the stricter StrongREJECT autograder, suggest that ETTA consistently induces harmful behavior across heterogeneous and stronger judgment methods, rather than merely exploiting quirks of a single evaluator.
>
> [2] JailbreakBench: An Open Robustness Benchmark for Jailbreaking Large Language Models, https://arxiv.org/abs/2404.01318
>
> [3] A StrongREJECT for Empty Jailbreaks, https://arxiv.org/abs/2402.10260
>
> ***5. More defense methods should be used in experiments to show that the proposed method can bypass existing defense methods.***
>
> We agree that broader defense coverage would strengthen our research. We have conducted an additional experiment with ReFAT [4]. ReFAT identifies “refusal feature” $r^{(l)}_{HH}$ for each layer by taking the mean difference between residual streams for harmful prompts and harmless prompts, and then performs adversarial training that ablates this direction while still forcing the model to output refusal responses.
>
> Following the original ReFAT paper, we implemented this defense on top of Llama-2-7b-chat-hf and Llama-3.2-3B-Instruct. Concretely, we estimate the refusal feature using 500 harmful prompts randomly sampled from AdvBench and 500 harmless prompts from Alpaca, and then apply the ReFAT loss with the same hyperparameters as reported in the original work. We then fine-tune the models and evaluate ETTA on JailbreakBench. The results are:
>
>    | Performance against Defense | Llama-2-7b-chat-hf | Llama-3.2-3B-Instruct |
>    | --------------------------- | ------------------ | --------------------- |
>    | CLEAN (no defense)          | 92.00%             | 95.00%                |
>    | ReFAT                       | 56.00%             | 70.00%                |
>
> ReFAT clearly reduces ETTA’s ASR, confirming that it is a strong baseline. However, ETTA still achieves 56–70% ASR. We hypothesize that this is because the ReFAT’s refusal feature $r^{(l)}\_{HH}$ is strongly tied to a specific training mixture (AdvBench + Alpaca). AdvBench harmful prompts are dominated by “classic” safety topics (e.g., weapons, physical harm), whereas JailbreakBench covers a more diverse set of harmful behaviors and contexts. The mean-difference estimator for $r^{(l)}\_{HH}$ is therefore biased toward the AdvBench distribution and may under-represent other types of harmfulness that appear in JailbreakBench. By contrast, ETTA performs word-level embedding perturbations and does not require fixing any particular harmful/harmless dataset pair to define a direction, which reduces this dataset-selection bias.
>
> [4]. Robust LLM safeguarding via refusal feature adversarial training, https://arxiv.org/abs/2409.20089
>
> ***6. More attack methods for white-box scenarios should be used for comparison.***
>
> We appreciate the concern about evaluation settings. Our intention in designing the evaluation suite was to cover diverse jailbreak paradigms. Concretely, our baselines include three white-box attacks: COLD-Attack (white-box automated jailbreak generation under multi-dimensional constraints), LLM Embedding Attack (optimization-based embedding poisoning in the continuous space),  Virus (a data-poisoning fine-tuning method with dual-goal optimization); and two black-box prompt-level iterative attacks PAIR and TAP . We fully understand that the current presentation may give the impression of an imbalance and adding more white-box baselines would make the comparison clearer and fairer. We have incorporated additional experiments against SCAV, a representative representation-engineering method. Here is a brief assessment result (we use 100 test cases, on three 32-layer victim LLMs ):
>
> ASR:
>
> | Method | Llama-2-7B | Llama-3-8B | Mistral-7B |
> | ------ | ---------- | ---------- | ---------- |
> | ETTA   | 89%        | 85%        | 91%        |
> | SCAV   | 90%        | 88%        | 89%        |

---

### Official Review · Reviewer_ANm1 · 2025-10-26

**Soundness:** 3
**Presentation:** 2
**Contribution:** 2
**Rating:** 4
**Confidence:** 5

**Summary:**

The paper proposes ETTA, an LLM attack framework. It can accurately identify and weaken the dimensions related to toxicity in the embedding vectors, and bring them below the model's safety detection threshold. ETTA does not require retraining the model and can deceive the model into executing malicious instructions while maintaining semantic coherence.

**Strengths:**

- This paper clearly empirically demonstrates and utilizes the linear separability of toxicity in the LLM embedding space and the threshold effect.
- The ASR achieved by ETTA is high, realistically illustrating the potential risks of open-source LLMs.
- The experiments in this paper are comprehensive, taking into account the scenarios of attacks (black-box and white-box).

**Weaknesses:**

- **This paper would significantly benefit from a more thorough comparison with related work**. The claim around L44 is unconvincing. Since 2023, several studies [1,2,3] have explored how internal representations affect LLM safety. While the proposed discovery and manipulation of toxicity at the embedding layer is relatively novel, the paper lacks strong empirical evidence demonstrating why attacks at the embedding level are more advantageous than those proposed by related work.
- **The assumptions regarding the attacker’s capacity are not well grounded**. Such assumptions should reflect realistic closed-source attack scenarios. In open-source settings, there is no reason to limit the attacker’s access, whereas in closed-source models, attackers are typically constrained to prompt manipulation rather than direct embedding modification.
- **ETTA relies on LLMs to detect semantic drift, which is cost expensive and inelegant**. While efficiency may not be the top priority for attacks, this approach introduces an unnecessary trade-off between quality and cost.
- **Sec 4.6 presents encouraging results on black-box attacks but lacks sufficient detail to be persuasive**. It remains unclear whether the embeddings used by OpenAI models directly correspond to their published embedding models, and how the proposed method extends to other closed-source models with undisclosed embedding models.

**Minor:**

The paper incorrectly applies citation formatting, where parenthetical references are incorrectly not bracketed. The authors should review the ICLR guidelines for proper citation usage.

[1] Representation Engineering: A Top-Down Approach to AI Transparency, https://arxiv.org/abs/2310.01405

[2] Uncovering Safety Risks of Large Language Models through Concept Activation Vector, https://arxiv.org/abs/2404.12038

[3] Refusal in Language Models Is Mediated by a Single Direction, https://arxiv.org/abs/2406.11717

**Questions:**

- The gray region in Fig. 2(b) contains several cases. Could authors characterize these boundary points? Do they reflect inherently ambiguous instances, or do they share properties with clearly classified cases but are affected by suboptimal boundary extraction?
- The threshold used in Fig. 2(b) appears loosely defined, as it depends on model-specific factors such as representation dimensionality. Could authors comment on whether this definition is meaningful and comparable across models?
- The claim that word embedding layer exhibits linear separability seems to conflict with prior work. For instance, [2] in weakness section reports that safety-related concepts typically emerge in middle layers, whereas early layers, especially layer 0, show negligible linear separability. I invite the authors to comment on this apparent discrepancy.

---

> ### Author Response · Authors · 2025-11-27
>
> We sincerely appreciate your time, careful reading of our work, and the insightful suggestions you provided!
>
> **Re for Weaknesses:**
>
> **W1&W2:**  We thank the reviewer for the thoughtful comments. We agree that the claim at L44 was overstated and will revise it to more accurately situate our work relative to recent studies. Regarding the attacker’s capacity, our threat model explicitly covers both white-box and closed-source scenarios. Our key contribution is to demonstrate that modifying only the initial token embeddings already suffices to bypass safety alignment, and to show how this embedding-level methodology can be extended to closed-source LLMs. The threat model is therefore centered on identifying and characterizing this relatively weak yet effective locus of control. In the revision, we will reorganize the threat-model discussion to clearly distinguish these two regimes and further articulate ETTA’s advantages and limitations in each setting.
>
> **W3:**  Regarding efficiency, ETTA uses a classifier LLM only during the offline search for a good embedding transformation; once this linear map is learned, each attack query incurs no additional classifier calls and only a single embedding-space transformation plus a normal forward pass. This design is aligned with a growing body of LLM safety work that treats efficiency cost as first-class concerns (e.g., [4] in ICLR25') and our results show that ETTA achieves strong ASR with a much lower per-query cost than most baselines. Moreover, our new experiments indicate that recent open-source LLMs can replace GPT-4-class judges with similar guidance quality, further reducing cost. We will clarify this amortization and cost profile more explicitly.
> | Classifier LLM        | Llama-2 | Llama-3 |
> | --------------------- | ------: | ------: |
> | ChatGPT-4o            |  87.88% |  84.81% |
> | Llama-3.2-3B-Instruct |  57.88% |  74.62% |
> | Qwen3-8B              |  83.85% |  79.04% |
>
> **W4:**  In Section 4.6, our method does not assume that the embeddings used by OpenAI’s chat models coincide exactly with their published embedding models; instead, we exploit the empirical observation that models in the same family tend to share a broadly compatible embedding geometry, so that a transformation learned on a surrogate (e.g., a related embedding endpoint or an open-source analogue) can transfer non-trivially to the API-only target. We will expand this section to describe more details.
>
> [4] Efficient Jailbreak Attack Sequences on Large Language Models via Multi-step Attack, https://openreview.net/pdf?id=jCDF7G3LpF

---

> ### Author Response · Authors · 2025-11-28
>
> **Re for Questions:**
>
> ***1. The gray region in Fig. 2(b) contains several cases. Could authors characterize these boundary points? Do they reflect inherently ambiguous instances, or do they share properties with clearly classified cases but are affected by suboptimal boundary extraction?***
>
> We appreciate the reviewer’s careful observation. In Fig. 2(b), the gray region collects the misclassified samples, i.e., all **false positives** and **false negatives**. Concretely, red points that fall below the decision boundary (x = 0) but should be classified as toxic are false negatives, while blue points that lie above the boundary but benign are false positives. Most of these boundary points correspond to words whose toxicity is strongly **context-dependent** and does not carry a clear intrinsic “harmful vs. benign” polarity at the word level, which only become clearly toxic or benign when placed in a full sentence. These ambiguous items therefore sit close to the decision boundary and are more easily flipped by small changes in the separating hyperplane. In this sense, they  highlight a genuine bottleneck of word-level classification, as struggle with words whose harmfulness is determined primarily by context.
>
> ***2. The threshold used in Fig. 2(b) appears loosely defined, as it depends on model-specific factors such as representation dimensionality. Could authors comment on whether this definition is meaningful and comparable across models?***
>
> We agree that the current definition of the gray-region threshold is not fully rigorous and is indeed model-specific. In the present version, we set the threshold \tau based on the signed distances of false positives and false negatives to the decision hyperplane (the farthest false negative red point has a signed distance of −0.0248, which is essentially aligned with the chosen cutoff $\tau = -0.025$). This makes the threshold meaningful within a given model, but not directly comparable across models, since the scale of distances depends on model-dependent factors.
>
> ***3. The gray region in Fig. 2(b) contains several cases. Could authors characterize these boundary points? Do they reflect inherently ambiguous instances, or do they share properties with clearly classified cases but are affected by suboptimal boundary extraction?***
>
> This performance difference stems from diverse experimental settings. In our experiments, we first apply PCA to project the token embedding vectors to a 50-dimensional subspace, in order to filter out noisy or low-variance directions and focus the classifier on the principal features. By contrast, [2] trains linear probes directly on the original high-dimensional activations without such dimensionality reduction. This should be the dominant variance in the experiment setup. Besides, although both works use a single hyperplane for classification, our probe is a linear SVM , whereas [2] employs a different optimization algorithm (as described in their paper).
>
> Thank you again for your thoughtful review, and please feel free to let us know if any further clarification or additional experiments would be helpful!

---

### Official Review · Reviewer_xjhK · 2025-10-31

**Soundness:** 2
**Presentation:** 2
**Contribution:** 2
**Rating:** 2
**Confidence:** 4

**Summary:**

The paper introduces ETTA, a framework that circumvents LLM safety alignment by manipulating embedding representations. The method is based on the finding that toxicity is linearly separable in the embedding space, allowing the authors to isolate and "attenuate" toxicity-related features via a linear transformation. This technique, guided by a classifier LLM, effectively bypasses model refusal mechanisms while maintaining model performance on standard benchmarks.

**Strengths:**

- The work is grounded in an empirical analysis that demonstrates a clear, linear separability between toxic and benign word embeddings, which forms the basis for the entire attack.
- The evaluation is comprehensive, testing the attack's effectiveness against five different open-source LLMs, its impact on general capabilities, and its resilience against several existing defense mechanisms

**Weaknesses:**

- The threat model assumes a high level of access, requiring the attacker to inject malicious code into the model's embedding pipeline to manipulate tensors at runtime. This is a more demanding prerequisite than many black-box or data-poisoning scenarios.
- The method's effectiveness and efficiency are heavily dependent on using a powerful, external classifier LLM (GPT-4o) to guide the attenuation factor search. The paper's own ablation study shows that substituting this with a smaller open-source model drastically reduces the attack success rate.
- The paper notes that the training of the linear transformation matrix is sensitive to random initialization, leading to a 12.4% variance in Attack Success Rate across different seeds. This suggests a degree of instability in reproducing the attack's optimal performance.
- What is the surprise part of this paper if we consider [1]?

[1] Representation Engineering: A Top-Down Approach to AI Transparency

**Questions:**

n/a

---

> ### Author Response · Authors · 2025-11-27
>
> We sincerely appreciate your time, careful reading of our work, and the insightful suggestions you provided!
>
> ***W1. The threat model assumes a high level of access.***
> We appreciate this concern and will clarify the scope of our threat model. Our setting follows a standard white-box / representation-editing threat model widely adopted in representation engineering attacks, where the attacker can perturb internal representations or embedding vectors at inference time. In fact, methods such as RepE [1] or our new baseline SCAV [2] assume the same access level, requiring the ability to manipulate the model’s full internal representation tensors.
>
> ***W2. The method's effectiveness and efficiency are heavily dependent on using a powerful, external classifier LLM (GPT-4o) to guide the attenuation factor search.***
> We appreciate this important point. The ablation in our paper uses Llama-3.2-3B-Instruct as the classifier LLM, which is weaker than GPT-4o and therefore leads to a drop in ASR.
> However, we want to emphasize that ETTA is not inherently tied to a closed-source classifier. Since open-source LLMs have been evolving rapidly, we have run new experiments using a recent, stronger open-source classifier LLM. The updated results show substantially improved guidance quality and attack performance, approaching the GPT-4o–based setting in our search procedure. This suggests that the external classifier can be replaced by a capable open-source alternative, which reduces cost and removes reliance on proprietary APIs, without changing the core ETTA mechanism.
> We will add this new experiment and discussion in the revision:
> | Classifier LLM         | Llama-2 | Llama-3 |
> | ---------------------- | ------: | ------: |
> | ChatGPT-4o             | 87.88%  | 84.81%  |
> | Llama-3.2-3B-Instruct  | 57.88%  | 74.62%  |
> | Qwen3-8B               | 83.85%  | 79.04%  |
>
> ***W3. The training of the linear transformation matrix is sensitive to random initialization.***
> Thank you for this observation. While seed sensitivity is indeed a limitation during the transformation-matrix training phase, we emphasize that this challenge is readily addressed in practical deployment scenarios. For any fixed victim model architecture, attackers can preserve the finalized linear transformation matrix post-optimization, thereby achieving deterministic inference outcomes. We will make this point explicit and report mean/variance across seeds more clearly.
>
> ***W4. What is the surprise part of this paper if we consider [1]***
> While ETTA shares conceptual similarities with representation steering, our key contribution lies in demonstrating that modifying **toxic-word-level token embeddings alone** is sufficient to circumvent safety alignment and induce systematic downstream refusal, which had not been shown in prior work. Unlike representation engineering methods like RepE, which actively steer hidden states or residual streams across multiple layers, ETTA operates by intervening solely on the token embedding prior to the first layer, without altering any intermediate representations. Moreover, RepE is primarily motivated by transparency and control and aims to better understand and control a model’s safety-related cognition and to improve harmlessness, rather than to analyze how safety alignment can be broken under constrained interventions.
>
> In light of the reviewer’s valuable suggestion, we conducted a preliminary comparison with a representative representation engineering attack method, SCAV. Here is a brief assessment result (we use 100 test cases, on three 32-layer victim LLMs ):
>
> ASR:
> | Method | Llama-2-7B | Llama-3-8B | Mistral-7B |
> | ------ | ---------- | ---------- | ---------- |
> | ETTA   | 89%        | 85%        | 91%        |
> | SCAV   | 90%        | 88%        | 89%        |
>
> Timecost(s):
> | Method | Llama-2-7B | Llama-3-8B | Mistral-7B |
> | ------ | ---------- | ---------- | ---------- |
> | ETTA   | 30.9       | 24.1       | 31.3       |
> | SCAV   | 113.84     | 94.32      | 102.18     |
>
> Results indicate that ETTA achieves comparable attack effectiveness while requiring substantially less intervention overhead. We will additionally include representative representation-engineering baselines in our final evaluation and further clarify these conceptual differences in the revised manuscript.
>
> [1] Representation Engineering: A Top-Down Approach to AI Transparency, https://arxiv.org/abs/2310.01405;
>
> [2] Uncovering Safety Risks of Large Language Models through Concept Activation Vector, https://arxiv.org/abs/2404.12038;
>
> Thank you again for your thoughtful review, and please feel free to let us know if any further clarification or additional experiments would be helpful!

---

### Official Review · Reviewer_tudL · 2025-11-02

**Soundness:** 3
**Presentation:** 3
**Contribution:** 2
**Rating:** 4
**Confidence:** 4

**Summary:**

This paper investigates embedding-space poisoning as a security threat to open-source LLMs and finds that toxic and benign prompts populate linearly separable regions of the embedding space. This reveals a geometric threshold that effectively governs refusal versus compliance. Leveraging this insight, the paper introduces ETTA, an embedding-level attack that first identifies toxicity-sensitive dimensions via a learned hyperplane, and then applies targeted linear transformations to attenuate those signals. ETTA bypasses refusals without fine-tuning and without access to training data, while preserving linguistic semantics.

On AdvBench and five popular open-source LLMs, ETTA achieves an average 88.61% attack success rate (+11.34 pp over the best baseline), generalizes to safety-enhanced and randomized-smoothing defenses (77.39% and 60.15% ASR, respectively), and induces only small capability drops on TruthfulQA and MMLU.

Contributions are: (i) empirically revealing a geometric vulnerability in current LLM safety alignment; (ii) proposing a practical embedding transformation that exploits this vulnerability to preserve coherence while reliably bypassing refusals.

**Strengths:**

1. This work reframes jailbreaks through the lens of embedding-space geometry: jailbreak triggers occupy linearly separable regions defined by toxicity-sensitive dimensions, which can be attenuated via simple linear transformations. Conceptually, this is an elegant synthesis of classical linear separability reasoning with modern LLM safety concerns, while practically removing major limitations of prior perturbation-based methods (e.g., optimization overheads and semantic drift).

2. The evaluation is thorough and well-designed: it spans five diverse open-source LLMs, employs a standard safety benchmark (AdvBench), reports concrete efficiency metrics (≈1.92 minutes per attack), and achieves state-of-the-art performance with an +11.34 percentage point ASR improvement over the best baseline. In addition, it includes stress tests against both instruction-tuned models and randomized smoothing defenses, and conducts careful utility assessments on TruthfulQA and MMLU, showing only modest degradation.

**Weaknesses:**

1. The paper does not engage with several state-of-the-art attack methods [1–3]. Given that ETTA is conceptually close to lines of work such as [2–4], a direct comparison is necessary to establish incremental value. Please add head-to-head results and a discussion clarifying what ETTA contributes beyond these approaches.

2. Because the study targets open-source LLMs, a straightforward “unlearning/guardrail removal” or lightweight model-editing baseline could provide a simpler pathway to bypass refusals. In addition, ETTA uses word-level replacements as part of its perturbation strategy for close models, which can induce semantic shifts and degrade relevance. Please quantify semantic drift (e.g., BERTScore/SimCSE to the original intent, factual consistency, human adequacy ratings) and report ASR without using the proposed method.

3. Commercial deployments typically combine model-internal guardrails with external moderation filters (e.g., [input/output classifiers](https://platform.openai.com/docs/guides/moderation)). The current evaluation focuses on model refusal but does not test end-to-end pipelines that include moderation. Please evaluate ETTA with realistic pre-/post-generation filters enabled, and report both (a) pre-moderation pass rates and (b) post-generation block rates. If ETTA’s gains diminish under these settings, discuss how to adapt the method (or why the failure is fundamental).

4. Because the proposed method bypasses guardrails via perturbations, please evaluate whether robustness/adversarial training (e.g., [5]) mitigates ETTA. Concretely, report (i) ASR against models trained with [5] under the same threat model (embedding-level perturbations), (ii) transferability of ETTA from standard to robustly trained models.


References:

[1] Andriushchenko et al. 2024. Jailbreaking Leading Safety-Aligned LLMs with Simple Adaptive Attacks

[2] Arditi et al. 2024. Refusal in Language Models Is Mediated by a Single Direction

[3] Wollschlager et al. 2025. The Geometry of Refusal in Large Language Models: Concept Cones and Representational Independence

[4]. Bayat et al. 205. Steering Large Language Model Activations in Sparse Spaces

[5]. Yu.  et al. 2024, Advanced defense: Robust llm safeguarding via refusal feature adversarial training

**Questions:**

Regarding the SVM training, did you train an individual SVM for each LLM?

---

> ### Author Response · Authors · 2025-11-29
>
> We sincerely thank the reviewer for the careful reading of our paper and for the thoughtful and constructive comments.
>
> **Re for Weaknesses:**
>
> ***W1. Add results and a discussion clarifying what ETTA contributes beyond representation engineering approaches.***
>
> We agree that ETTA is conceptually related to representation steering. We have now made this connection explicit and clarified that our contribution is not to introduce an entirely new paradigm, but to show that **modifying only word-level token embedding weights** (before the first layer) suffices to bypass safety-alignment, which was not fully demonstrated in prior works [1-3] that exclusively edits layer-wise hidden states/residual streams we remain all intermediate representations unchanged). By preserving all intermediate representations, this lightweight pre-layer perturbation reliably induces systematic refusal/jailbreak behavior, revealing a distinct locus of control.
>
> However, we agree the reviewer's constructive opinion on appending more representation engineering attacks as evaluation baselines, so we test several representative works for a more comprehensive evaluation. Here is a brief assessment result (we use 100 test cases, on three 32-layer victim LLMs):
>
> ASR:
>
> | Method | Llama-2-7B | Llama-3-8B | Mistral-7B |
> | ------ | ---------- | ---------- | ---------- |
> | ETTA   | 89%        | 85%        | 91%        |
> | SCAV   | 90%        | 88%        | 89%        |
>
> Timecost(s):
>
> | Method | Llama-2-7B | Llama-3-8B | Mistral-7B |
> | ------ | ---------- | ---------- | ---------- |
> | ETTA   | 30.9       | 24.1       | 31.3       |
> | SCAV   | 113.84     | 94.32      | 102.18     |
>
> The comparative analysis reveals that while ETTA and SCAV exhibit equivalent ASR, ETTA demonstrates enhanced operational efficiency, which highlights ETTA's potential for real-time applications. We will add evaluations against more representative baselines and report full results in the revision.
>
> ***W2. Quantify semantic drift and report ASR without using the proposed method.***
>
> We thank the reviewer for raising the concern that a straightforward model-editing baseline could also provide a simpler pathway to bypass refusals. Our ETTA framework is designed to achieve jailbreaking with substantially lower overhead. In contrast, approaches that directly modify model weights, such as Virus, the finetuning-based data poisoning baseline included in our experiments, typically incur large memory and training-time costs. And low-cost model-editing can significantly degrade the model’s normal performance on benign inputs. Our core contribution is to show that **modifying only the word-level token embedding weights** already suffices to bypass safety alignment, while avoiding the heavy computational and performance overheads introduced by full model retraining or aggressive guardrail removal.
>
> To address the reviewer’s concern about semantic shifts introduced by word-level replacements (introduced in Section 4.6), we explicitly measure semantic similarity between the original harmful instruction P and the transformed prompt P′ actually sent to the target LLM. Concretely, for a random chosen subset of 100 prompt pairs ⟨P, P′⟩, we compute **BERTScore-F1** with a pre-trained RoBERTa encoder. The average BERTScore is 0.94 cross all attacked prompts, indicating that our toxic-word-targeted perturbations preserve the original intent with minimal semantic changes.
>
> The reviewer correctly points out that sometimes an unmodified jailbreak prompt may already bypass the model’s defense. This phenomenon has been explored in prior work; for example, the original AdvBench paper reports non-zero ASR on closed-source models when directly using the harmful prompts, including **1.8% on GPT-3.5** and **8.0% on GPT-4** [6]. Following this insight, we add an **Original ASR** baseline, where each target LLM is directly queried with the dataset prompts **without** any attack method or transformation, and we measure the attack success rate under the same success criterion as in our main experiments. The results are:
> | Target Model          | None-Jailbreak |
> | --------------------- | -------------- |
> | GPT-4o                | 0.00%          |
> | Llama-2-7b-chat-hf    | 2.12%          |
> | Llama-3.2-3B-Instruct | 13.65%         |
> | Qwen2.5-7B-Instruct   | 10.96%         |
> | vicuna-7b-v1.5        | 5.58%          |
> | gemma-2-9b-it         | 3.85%          |
>
> [6] Universal and Transferable Adversarial Attacks on Aligned Language Models, https://arxiv.org/html/2307.15043.

---

> ### Author Response · Authors · 2025-11-29
>
> ***W3. Against external moderation filters***
>
> This is a valuable request. Our current evaluation of ETTA is intentionally restricted to model-internal safety alignment, which matches the most common setting in research and many open-source deployments and provides a clean, comparable testbed. While we agree that commercial systems typically combine such internal guardrails with additional input/output moderation, these pipelines vary substantially across providers. Committing to any particular pipeline would introduce many deployment-specific design choices that are orthogonal to the core contribution of ETTA. Conceptually, however, ETTA is compatible with moderated deployments: it can be safely combined with existing guardrail-bypass techniques (e.g., [7]) on the input side. We view this suggestion an important direction for future work.
>
> [7] Jailbreaking Large Language Models Against Moderation Guardrails via Cipher Characters, https://arxiv.org/abs/2405.20413.
>
> ***W4. Report (i) ASR against models trained with [5] under the same threat model, (ii) transferability of ETTA from standard to robustly trained models.***
>
> We thank the reviewer for this insightful suggestion and for pointing us to [5]. We appreciate the reviewer’s expertise in this area. We agree that ReFAT [5] is highly related to our setting, since it explicitly perturbs internal representations along a “refusal feature” direction to improve robustness.
>
> Following the reviewer’s suggestion, we implemented ReFAT on top of our base models and evaluated ETTA under the same embedding-level threat model as in the main paper. Concretely, we closely followed the methodology and hyperparameters in [5]: for each layer, we estimated the refusal feature $r^{(l)}_{HH}$ as the mean difference between harmful and harmless prompts, using 500 harmful prompts from AdvBench and 500 harmless prompts from Alpaca. After ReFAT training, we attacked the resulting defended models with ETTA and evaluated on JailbreakBench. The results are:
>
>    | Performance against Defense | Llama-2-7b-chat-hf | Llama-3.2-3B-Instruct |
>    | --------------------------- | ------------------ | --------------------- |
>    | CLEAN (no defense)          | 92.00%             | 95.00%                |
>    | ReFAT                       | 56.00%             | 70.00%                |
>
> ReFAT clearly reduces ETTA’s ASR, but does not eliminate it: even after adversarial training, ETTA still achieves 56–70% ASR on JailbreakBench. We hypothesize that this is because the ReFAT defense is based on a mean-difference vector ($r^{(l)}_{HH}$), which only captures a single dominant harmful–harmless axis estimated from a specific training mixture (AdvBench + Alpaca). In contrast, ETTA learns a higher-rank transformation matrix in the embedding space, which can analysis more complex directions in the representation geometry (at the cost of reduced interpretability).
>
>    Moreover, ReFAT’s refusal feature $r^{(l)}\_{HH}$ is strongly tied to a specific training mixture (AdvBench + Alpaca). AdvBench harmful prompts are dominated by “classic” safety topics (e.g., weapons, physical harm), whereas JailbreakBench covers a more diverse set of harmful behaviors and contexts. The mean-difference estimator for $r^{(l)}\_{HH}$ is therefore biased toward the AdvBench distribution and may under-represent other types of harmfulness that appear in JailbreakBench. By contrast, ETTA performs word-level embedding perturbations and does not require fixing any particular harmful/harmless dataset pair to define a direction, which reduces this dataset-selection bias.
>
>    Regarding “robustly trained models”, we understand this as including both adversarially trained models such as ReFAT [5] and safety-tuned models obtained via instruction tuning. Our main paper already evaluates ETTA against another robustness-oriented defense, **Enhanced Safety Finetuning (ESF)** (Safety-aware instruction tuning with minimal examples). There we observe only an **11.24% average ASR drop** when applying ETTA to ESF-tuned models compared to their standard counterparts. This is consistent with our finding: model rejection behavior follows a geometric threshold effect where safety finetuning also reshapes decision boundaries in embedding space. Because ETTA operates directly in this embedding space, its learned perturbations tend to transfer across such robustly trained models. If our understanding of “robustly trained models” is not fully aligned with the your intent, we would be happy to refine our experiments and clarifications further.

---

> ### Author Response · Authors · 2025-11-29
>
> **Re for Questions:**
>
> ***Q1. Regarding the SVM training, did you train an individual SVM for each LLM?***
>
> The answer is Yes, we do need to train a separate SVM for each target LLM. This is necessary because different LLMs exhibit distinct token-embedding spaces, even when they are architecturally related or obtained via tuning from a common base model. For example, models such as Mistral and Llama2 (or a tuned variant versus its initialization) do not share an aligned embedding geometry; therefore, a classifier learned on one embedding space does not transfer reliably to another. Training an individual SVM per LLM ensures that the decision boundary is correctly defined in that model’s native embedding manifold. We will clarify this in the revision.
>
> We appreciate your time and consideration in reviewing our work, and we would be happy to further clarify any remaining questions or concerns.

---

### Author Response · Authors · 2025-12-02

Dear Area Chair,

We have addressed all reviewers' questions and concerns in our rebuttal and posted the requested experimental results. Also, the revised paper contains all the required changes.  Due to the unexpected circumstance, we were unable to receive further feedback from the reviewers during the rebuttal period. Therefore, we would like to take this opportunity to briefly summarize the key points from the review phase regarding our submission.

In summary, there was broad agreement on the main strength of our work: ETTA is grounded in an empirical analysis showing a clear linear separability between toxic and benign word embeddings, and then leverages this structure by modifying the initial token embedding weights to bypass safety alignment. We are grateful for the reviewers’ careful reading and constructive feedback.

The core concerns raised by the reviewers focus on four points, and we addressed each directly in our rebuttal. (i) First, regarding **the relationship to prior representation-engineering work**, we clarified that our contribution is to show that a minimal pre-layer embedding intervention suffices to break alignment and added a head-to-head comparison with SCAV, where ETTA achieves comparable ASR with significantly lower time cost. (ii) Second, on the **realism of the threat model**, we reorganized the discussion to clearly distinguish white-box/open-source and closed-source settings, emphasizing that our primary focus is on open-source/self-hosted deployments and framing the black-box experiments as a constrained proof-of-concept via exposed embedding interfaces. (iii) Third, to address concerns about **reliance on classifier GPT-4o**, we reported new results with a strong open-source classifier LLM that closely matches GPT-4o’s behavior in our pipeline, thereby reducing both cost and dependence on proprietary APIs. (iv) Finally, with respect to **evaluation breadth**, we reported new results on expanding defenses and harmfulness-judging methods to demonstrate the consistency and reliability of our evaluation.

In the final revision, guided by these comments, we have incorporated the SCAV baseline into our evaluation, refined the introduction and threat-model sections to more accurately situate ETTA, and expanded the discussion of classifier LLM choices and their implications for efficiency and applicability.

Thanks for your patience. We truly appreciate your time and effort throughout the process. If you have any concerns or questions regarding our response, please feel free to let us know.

Sincerely,
The authors

---

### Meta-Review · Area_Chair_QEis · 2026-01-17

**Summary:**

This paper proposes ETTA, a method for breaking LLM safety alignment by manipulating input embeddings. The paper is clearly written and presents extensive experiments across multiple models. Several reviewers find the empirical observation of linear separability in embedding space to be interesting and potentially insightful.

However, despite these strengths, multiple reviewers consistently raise concerns about the threat model, the incremental contribution relative to prior work, and the evaluation of the proposed method. After carefully considering both the original submission and the rebuttal, I believe these issues remain substantial and prevent the paper from meeting the acceptance bar at this time.

**Reviewer Concerns:**

* Reviewers xjhK, ANm1, and h6QN raised a fundamental concern about the threat model, where the attacker is assumed to be able to inject malicious code into the model's embedding pipeline to manipulate tensors at inference. Such assumptions do not reflect realistic closed-source attack scenarios, and in open-source settings, there is no reason to limit the attacker's access. While the rebuttal frames the main contribution as identifying a "weak locus of control" (token embeddings alone suffice to break refusal), it is questionable whether restricting the attacker to the input embedding layer is meaningfully different from allowing broader internal activation edits: in practice, both may be similarly feasible (or infeasible) depending on the deployment, so the proposed threat model may not clearly correspond to a distinct real-world security boundary.

* Reviewers tudL, ANm1 and h6QN questioned the paper's incremental value relative to prior representation-engineering works, such as SCAV. While the rebuttal adds head-to-head experiments, the results do not establish a clear advantage: the proposed method does not outperform SCAV in attack success rate, and it also does not consistently outperform Embedding Attack in terms of time efficiency.

* Reviewers tudL and h6QN raised concerns about whether the model responses induced by the attack are genuinely harmful and whether the reported attack success rates are inflated by the semantic drift in input prompts or quality decline in the model's general capabilities. While the rebuttal reports semantic similarity metrics and comparisons across automated judges, it does not provide direct evidence (e.g., human evaluation) that the generated outputs consistently contain substantive harmful guidance rather than superficial relevance. Moreover, it is worth noting that the LLM evaluator used in the paper primarily checks relevance to the query, rather than explicitly checking whether the response provides genuinely useful information for harmful queries.

* Reviewer xjhK and ANm1 raised concerns about the method's reliance on a strong external classifier LLM (GPT-4o) to guide the attack. The authors added new experiments to show that small open-source models are also reliable enough to induce high attack success rates.

**Reviewer Scores:**

* tudL is likely to maintain a score of 4 because the current results do not establish a clear advantage over SOTA methods.
* xjhK may raise the score to 4 due to the authors' additional experiments that have addressed the concern about using external LLMs, though the concern about the threat model remains.
* ANm1 is likely to maintain a score of 4 because of the concerns on the threat model and comparison with related works.
* h6QN is likely to maintain a score of 6 because several of their concerns are not fully addressed.

---

### Decision · Program_Chairs · 2026-01-26

Reject